# Haematochemical Profile of Healthy Dogs Seropositive for Single or Multiple Vector-Borne Pathogens

**DOI:** 10.3390/vetsci11050205

**Published:** 2024-05-08

**Authors:** Raffaella Cocco, Sara Sechi, Maria Rizzo, Andrea Bonomo, Francesca Arfuso, Elisabetta Giudice

**Affiliations:** 1Department of Veterinary Sciences, Teaching Veterinary Hospital, University of Sassari, Via Vienna 2, 07100 Sassari, Italy; rafco@uniss.it (R.C.); sarasechilavoro@tiscali.it (S.S.); 2Department of Veterinary Sciences, University of Messina, Polo Universitario dell’Annunziata, 98168 Messina, Italy; anbonomo@unime.it (A.B.); farfuso@unime.it (F.A.); egiudice@unime.it (E.G.)

**Keywords:** vector-borne diseases, dog, haematochemical parameters, endemic area

## Abstract

**Simple Summary:**

Vector-borne diseases (VBDs) encompass illnesses transmitted by various pathogens through arthropod vectors, such as ticks, fleas, mosquitoes, and sand flies. These diseases significantly affect human and veterinary health globally due to the close relationship between domestic animals and humans and the zoonotic potential of these pathogens. This study focuses on the antibody response to common VBDs (i.e., *Ehrlichia canis, Rickettsia* spp*., Leishmania* spp., *Borrelia burgdorferi*, *Anaplasma phagocytophilum*, and *Bartonella henselae*) and its impact on the haematochemical profile of healthy dogs in Sardinia, Italy. This study involves the laboratory analysis of biochemical profiles and serological tests for various pathogens. The results show different haematochemical patterns in dogs with single and multiple pathogen exposures, indicating potential liver and kidney damage and inflammatory responses. However, further confirmation through molecular or parasitological diagnostics is necessary. This study highlights the importance of understanding co-infections and their effects on canine health in VBD-endemic regions, suggesting the need for further research in this area.

**Abstract:**

Background: the present study aimed to investigate the immunological response to common vector-borne pathogens and to evaluate their impact on haematochemical parameters in owned dogs. Methods: Blood samples were collected from 400 clinically healthy dogs living in an endemic area (Sardinia Island, Italy). All dogs were serologically tested for VBDs and divided into groups based on their negative (Neg) or positive response towards *Ehrlichia* (Ehrl), *Rickettsia* (Rick), *Leishmania* (Leish), *Borrelia* (Borr), *Anaplasma* (Anapl), and *Bartonella* (Barto). A Kruskall–Wallis’s test, followed by Dunn’s post hoc comparison test, was applied to determine the statistical effect of negativity and single or multiple positivity on the studied parameters. Results: the group of dogs simultaneously presenting antibodies towards *Leishmania, Ehrlichia*, and *Rickettsia* showed higher values of total proteins, globulins, creatine phosphokinase, aspartate aminotransferase, alanine aminotransferase, and amylase than dogs that tested negative or dogs with antibodies toward a single pathogen investigated herein. Conclusions: Our results seem to suggest that exposure to more vector-borne pathogens could lead to greater liver function impairment and a greater inflammatory state. Further investigations are needed in order to better clarify how co-infections affect haematochemical patterns in dogs living in endemic areas of VBDs.

## 1. Introduction

Vector-borne diseases (VBDs) are a group of illnesses caused by a range of pathogens, including *Ehrlichia canis*, *Rickettsia* spp., *Leishmania* spp., *Borrelia burgdorferi*, *Anaplasma phagocytophilum*, and *Bartonella henselae*. These pathogens are transmitted to dogs through the blood meal of arthropods acting as vectors (i.e., ticks, fleas, mosquitoes, and sand flies). These vectors need protein components as nutrients for their gonadotrophic cycles. The requirement for a blood meal from a host to generate egg batches underscores the obligate parasitic lifestyle that certain arthropods have adopted, resulting in the laying of a notably greater quantity of eggs compared to unfed individuals [1]. In addition to transmission via ticks, pathogens can also be disseminated through other means by arthropods, such as depositing pathogens in their faeces, transmission through the ingestion of contaminated material, or when arthropods feed on ocular secretions [1]. Hence, the intricate networks of vector-borne transmission pathways stand as profoundly captivating and intricate instances of evolution and interactions among pathogens, hosts, and vectors, shaped under the influence of various ecological and environmental factors [1]. The duration of pathogen transmission is governed by various biological variables related to vectors, pathogens, and host immune responses [2]. VBDs significantly impact human health, contributing to approximately one million deaths annually [3,4]. Their relevance in veterinary medicine is underscored by the increasingly close relationship between domestic animals and humans globally [1], coupled with their zoonotic potential [5,6], which allows them to infect animals in diverse geographical and socio-economic contexts. Public desire to adopt dogs from abroad that have often had their welfare compromised by events, such as natural disasters, is increasing. In part, this is driven by social media channels in affluent regions and increased awareness of geographically distant homeless dogs. Consequently, dogs are often relocated across extensive geographical areas. Both stray and owned dogs, if not properly treated with endo- and ectoparasiticides, are at a heightened risk of exposure to vector-borne pathogens, potentially serving as competent reservoirs [7,8]. Additionally, climate exerts a significant influence, affecting the life cycles of many infectious agents [9,10,11]. In dogs, ehrlichiosis, rickettsiosis, leishmaniosis, borreliosis, anaplasmosis, and bartonellosis are the most prevalent, and sometimes fatal, vector-borne diseases. Generally, the Mediterranean basin provides an ideal environment for the circulation of VBDs in domestic animals. Ongoing monitoring of local canine populations and the maintenance of updated epidemiological data are crucial due to limited available information, often restricted to specific countries or pathogens [12,13,14]. Notably, Sardinia Island (Italy) has long been endemic for leishmaniasis, with recent reports indicating the presence of two arthropod vectors in the region [15]. Sardinia is the second biggest island in the Mediterranean Sea, located approximately halfway between Spain, Italy, and North Africa. The Sardinian climate is typically Mediterranean, with an annual mean temperature of 22 °C, which allows the survival of many arthropod vectors throughout the whole year. Furthermore, the island is an important stopover area for migratory birds, posing a risk for the introduction and dispersal of ticks and TBDs [16]. The present study aimed to evaluate the antibody response towards the most common vector-borne pathogens (*Ehrlichia*, *Rickettsia*, *Leishmania*, *Borrelia*, *Anaplasma*, and *Bartonella*) and its impact on the haematochemical profile of clinically healthy owned dogs living in Sardinia.

## 2. Materials and Methods

### 2.1. Animals

This study involved a retrospective analysis of a population of dogs who presented at the Veterinary Teaching Hospital (VTH) of the University of Sassari (Sardinia, Italy) for routine screening and prophylaxis. Dogs were included in the study if they exhibited the following criteria: their owners provided informed consent for the scientific use of their animal’s data; they were clinically healthy upon physical examination, free from external and internal parasites, and in good nutritional condition; their biochemical profile was checked at the time of hospital admission; and at least one aliquot of serum was stored frozen at −20 °C.

Dogs were excluded if they exhibited the following criteria: they showed physical or historical signs of any kind; they were undergoing any pharmacological treatment, including preventative ectoparasite treatment in the month prior to blood sampling; they had a history of vector-borne disease; and they were vaccinated against the investigated diseases (i.e., leishmaniosis and borreliosis).

### 2.2. Laboratory Analysis

The chemistry profiles obtained upon hospital admission were reviewed for all enrolled dogs. Total Proteins (TP), Albumin (Alb), Globulins (Glob), Glucose (Glu), Triglycerides (TG), Amylase (AMYL), Lipase (LPS), Cholesterol (Chol), Alkaline Phosphatase (ALP), Creatine phosphokinase (CPK), Urea, Creatinine (Crea), Total Bilirubin (TB), Gamma-Glutamyl Transferase (GGT), Aspartate Aminotransferase (AST), Alanine Aminotransferase (ALT), Calcium (Ca), and Phosphorus (P) were detected using a commercial kit, according to the manufacturer’s instructions, and an automated UV Spectrophotometer (SEAC, Slim, Florence, Italy). All testing was conducted within the laboratory at the Department of Veterinary Medicine at the University of Sassari.

The serum samples collected at the time of hospital admission, after thawing at room temperature, were tested for *Ehrlichia*, *Rickettsia*, *Leishmania*, *Borrelia*, *Anaplasma*, and *Bartonella* antibodies.

Anti-*Ehrlichia canis* antibodies were detected by IFAT using slides containing fixed *E. canis* antigen (ATCC no. CRL10390) prepared in monocyte–macrophage cells (DH82), as described by Dawson et al. [17]. Briefly, infected cultures were split 1:2, were layered onto normal DH82 cells, and were permitted to grow in 25 cm 2T-flasks (Corning, NY, USA) containing Earle’s salts Minimal Essential Medium (E/MEM, Gibco, NY, USA) supplemented with 2 mM l-glutamine (200 mM) and 15% (*v*/*v*) fetal calf serum (FCS, Gibco) at 37 °C with 5% CO_2_. After about a month, *E. canis*-infected DH82 cultures were harvested using centrifugation at 800× *g* for 10 min and were washed twice with phosphate-buffered saline (PBS, 0.1 M phosphate, 0.33 M NaCl, pH 7.2). The final pellets were resuspended in PBS and were dropped (10 µL) into each well of a 12-well Teflon-coated slide (Immuno-Cell Int.). The slides were air-dried for 1 h, fixed with cold acetone for 15 min, and were used immediately or stored at −20 °C until use. A titre of at least 1:80 was considered positive, as this is diagnostic of ehrlichiosis due to the persistent nature of this organism and the high seroprevalence in Sardinia [18].

To detect anti-*Rickettsia* spp. antibodies, a commercial canine *Rickettsia rickettsia* IgG Indirect Fluorescent Antibody Test (IFAT) KIT (Rickettsia IFA, Fuller Laboratories, Fullerton, CA, USA) was used. A positivity threshold of at least 1:128 was established to exclude low antibody titres, which could potentially result from nonspecific fluorescence. This threshold was chosen because infected dogs typically develop this antibody titre by the time clinical signs appear and during the initial sample collection [19,20]. In a comparative study assessing various methods for the serological diagnosis of spotted fever, IFAT demonstrated a sensitivity index of 94%, with positive results considered at titres greater than 1:128 [20].

Anti-*Leishmania infantum* IgG antibodies were detected using an in-house IFAT following the laboratory procedures outlined in the OIE Manual of Diagnostic Tests and Vaccines for Terrestrial Animals [21]. Promastigotes of *L. infantum* zymodeme MON-1 served as the antigen, with dilutions starting from 1:40. Positive control serum from a confirmed infected dog was included. The samples were deemed positive for Canine Leishmaniasis when they exhibited clear cytoplasmic and membrane fluorescence for promastigotes at a dilution of 1:80, following the guidelines of the Italian National Reference Centre for Leishmaniasis (C. Re. Na. L.—Istituto Zooprofilattico di Palermo, Palermo, Italy), as detailed by Foglia Manzillo et al. [22]. Positive sera were titrated until negative results were obtained. The highest dilution at which fluorescent promastigotes were observed determined the antibody titre, whereas samples showing fluorescence only at a 1:40 dilution were considered exposed but not infected.

Antibodies against *Borrelia burgdorferi* were detected using the commercial Canine Borreliosis IgG IFA test (FULLER Laboratories, Fullerton, CA, USA), following the manufacturer’s instructions. Serum samples were diluted (1:40, 1:80, 1:160, 1:320, 1:640, and 1:1280) in phosphate-buffered saline, with 20 mL of each dilution transferred to Lyme antigen-coated wells on commercially purchased slides. Each slide included both positive and negative control sera, with 2 samples tested per slide. The slides were then placed in a humid chamber at 37 °C for 30 min, followed by a 10-min soak in PBS. After rinsing with distilled water and drying, FITC-labelled anti-equine IgG (H+L) antibody (KPL, Gaithersburg, MD, USA) was added at a 1:40 dilution, and the slides were re-incubated for 30 min at 37 °C. Following another wash, the slides were overlaid with one drop of glycerol-based fluorescent mounting medium (KPL, Gaithersburg, MD, USA). A cover slip was applied, and the slides were visualized using standard fluorescence microscopy. A positive reaction was indicated by sharply defined apple-green fluorescence, akin to the negative control well. All sera with titres ≥1:64 were considered positive.

Slides containing *Anaplasma phagocytophilum* antigen (Fuller Laboratories) were used for detecting antibodies against *A. phagocytophilum*. A titre of at least 1:40 was considered positive, since references concerning this infection in dogs are scarce in Sardinia. The IFAT results were expressed as positive, negative, or doubtful. Doubtful sera were used to indicate an intermediate zone, a term used synonymously for a zone of test values between the positive and negative cut-offs that can vary depending on the cut-off chosen for the test [21].

From the other serum aliquots, an indirect immunofluorescence test was performed to detect IgG and IgM against *Bartonella henselae* using slides containing the antigen *Bartonella henselae* (Houston 1 ATCC 49882) cultured in the L929 fibroblast cell line. A serum dilution of 1:40 was used as a cut-off for both classes of antibodies.

### 2.3. Statistical Analysis

A descriptive statistical analysis was applied to evaluate the characteristics of the data set distribution, including gender, age, seronegativity, and simple or multiple seropositivity towards vector borne diseases. All data were tested for normality of distribution using a Shapiro–Wilk test. All data were non-normally distributed (*p* < 0.05). A Kruskall–Wallis’s test, followed by Dunn’s post hoc comparison test, was applied to determine the statistical effect of negative or positive status on the haematochemical parameters studied. A *p* value of <0.05 was considered statistically significant. Statistical analyses were performed using the statistical software R v. 4.3.0 (R Core Team, Vienna, Austria, 2023).

## 3. Results

For this study, 400 dogs (median age: 5 years, range 1–18 years; 218 males, 182 females) met the inclusion and exclusion criteria (Table 1). All the animals were kept as pets.

The study population of dogs was divided into groups according to their response to serological testing: negativity (Neg), simple positivity for *Ehrlichia* (Ehrl), *Rickettsia* (Rick), *Leishmania* (Leish), *Borrelia* (Borr), *Anaplasma* (Anapl), and *Bartonella* (Barto), or multiple positivity (Table 2).

Regarding the antibody response, the descriptive statistical analysis revealed 35 seronegative dogs (Neg; 8.75%), 200 simple seropositive dogs, in particular 57 *Ehrlichia*- (Ehrl; 14.25%), 75 *Rickettsia*- (Rick; 18.75%), 51 *Leishmania*- (Leish; 12.75%), 1 *Anaplasma*- (Anapl; 0.25%), and 16 *Bartonella*-positive (Barto; 4.00%) dogs. No dogs were titred positive towards *Borreli*a alone (Borr; 0.00%). Multiple seropositive dogs (165; 41.25%) were split in to subcategories depending on the antibody response, as follows: 4 Leish/Barto group, 1 Leish/Borr, 7 Rick/Barto, 19 Rick/Leish, 4 Rick/Leish/Barto, 16 Ehrl/Barto, 17 Ehrl/Leish, 5 Ehrl/Leish/Barto, 56 Ehrl/Rick, 13 Ehrl/Rick/Barto, 17 Ehrl/Rick/Leish, 4 Ehrl/Rick/Leish/Barto, 1 Ehrl/Rick/Leish/Anapl, and 1 Ehrl/Rick/Leish. The Kruskall–Wallis’s test showed lower TP values in the N group with respect to the C and Ehrl/Rick/Leish groups; however, the TP values were higher in the Ehrl/Rick/Leish group compared to the Ehrl, Rick, Barto, and Ehrl/Rick groups. A higher concentration of Alb was found in the Neg group compared to the Ehrl, Rick, Leish, Ehrl/Rick, Ehrl/Rick/Leish, Ehrl/Rick/Leish/Barto, Ehrl/Rick/Barto, Ehrl/Leish, Rick/Leish, and Rick/Leish/Barto groups. A lower concentration of Glob was found in the Neg group compared to the Ehrl, Rick, Leish, Ehrl/Rick, Ehrl/Rick/Leish, Ehrl/Rick/Barto, Ehrl/Leish, Rick/Leish, and Rick/Leish/Barto groups, whereas it was higher in the Ehrl/Rick/Leish group compared to the Ehrl, Rick, and Barto groups. No statistical significance was found in Crea concentration among the groups. The urea values were lower in the Neg group compared to the Ehrl, Rick, Leish, Barto, Ehrl/Rick, Ehrl/Rick/Leish, Ehrl/Leish, Ehrl/Leish/Barto, and Rick/Leish groups (Figure 1).

A lower ALP concentration was found in the Neg group with respect to the Leish, Ehrl/Rick/Leish, and Ehrl/Leish groups. The GGT values were lower in the Neg group compared to the Ehrl, Rick, Leish, Barto, Ehrl/Rick, Ehrl/Rick/Leish, Ehrl/Leish, and Rick/Leish groups. A lower concentration of CPK was found in the Neg group compared to the Ehrl, Leish, Barto, Ehrl/Rick, Ehrl/Rick/Leish, Ehrl/Leish, and Ehrl/Leish/Barto groups, and in the Rick group compared to the Ehrl/Rick/Leish group. The TB values were higher in the Neg and Ehrl groups compared to the Rick group. The AST concentrations were lower in the Ehrl/Rick group compared to the Neg, Ehrn, Rick, Leish, Barto, Ehrl/Rick/Leish, Ehrl/Rick/Barto, Ehrl/Leish, Ehrl/Leish/Barto, Rick/Leish, and Rick/Leish/Barto groups, whereas the AST concentrations were higher in the Ehrl/Rick/Leish group compared to the Neg and Ehrl groups. A higher concentration of ALT was found in the Ehrl/Rick/Leish group compared to the Ehrl/Rick group. The AMYL concentrations were lower in the Neg group compared to the Rick, Leish, Ehrl/Rick, Ehrl/Rick/Leish, Ehrl/Rick/Barto, Ehrl/Leish, and Rick/Leish groups, and in the Ehrl group compared to the Ehrl/Rick/Leish group. Lower LPS concentrations were found in the Neg group compared to the Ehrl, Rick, Leish, Barto, Ehrl/Rick, Ehrl/Rick/Leish, Ehrl/Rick/Barto, Ehrl/Leish, and Rick/Leish group. The Ca values were lower in the Barto group compared to the Neg group (Figure 2). No significance differences were found in P, Glu, Chol, and TG concentrations between the groups.

## 4. Discussion

The determination of haematochemical parameters has emerged as a crucial clinical diagnostic tool. As a matter of fact, changes in the concentration of these blood parameters are related to organic abnormalities associated with various diseases, and they can serve as a foundation for further comprehensive studies. Over the years, extensive research has been conducted on haematochemical parameters in domestic animals, including cattle, dogs, cats, and sheep. However, some parameters have not received enough attention in the context of healthy states versus exposed states.

In this study, the haematochemical profiles of clinically healthy animals with or without serum antibodies towards the main vector-borne pathogens, either alone or in combination with each other, were analysed.

Our results demonstrate a slight increase in TP. In bartonellosis, the haematochemical profiles and urinalysis findings frequently appear normal. Nonetheless, reported laboratory abnormalities include anaemia, eosinophilia, hyperproteinaemia, hyperglobulinaemia, neutropenia, and thrombocytopenia [23]. This pathogen is also involved in the development of infective endocarditis [24], but our group showed no sign of heart-related analytes that may indicate heart failure.

In this study, dogs with antibodies against *E. canis* showed normal levels of ALB compared to the control group. This suggests that renal damage, typically associated with ehrlichiosis pathogenesis, may not have been severe, and that renal function remained within normal limits. This is further supported by the normal levels of Crea, despite slightly elevated levels of P and urea. Conversely, ALP levels appeared to be higher. The enzymatic activity of ALP was found in hepatocytes, the heart, and skeletal muscle cells, indicating potential damage or lesions in these tissues, which could account for its increase in the blood. Additionally, CPK levels were doubled compared to the negative group, reinforcing the likelihood of damage to muscular tissues [25]. Although there is no clear evidence in the literature linking pancreatic damage to ehrlichiosis, dogs may exhibit higher values of AMYL, LPS, and TG without displaying clinical signs of pancreatitis [26]. However, LPS and AMYL are not specific for the pancreatic damage [27]. Regarding TB and GGT, ehrlichiosis can lead to interruptions in biliary flow [25], explaining the elevated values of TB and GGT found in infected dogs. Furthermore, *Ehrlichia*-induced hepatic damage causes decreased protein production by the liver and is associated with increased levels of AST and ALT.

In dogs infected with *Rickettsia* spp., ALP, AST, and ALT levels were higher compared to the control group, aligning with the existing literature. TB values appeared slightly elevated, whereas GGT levels were higher, suggesting potential liver damage. Kidney analytes showed no significant difference from the control group, except for slightly elevated urea levels. CPK, AMYL, and LPS levels were remarkably high in this group, warranting further investigation into the underlying causes.

It is well known that alterations in urinalysis and haematochemical evaluations often reflect liver or kidney involvement due to *Leishmania* multiplication within macrophages in the liver, leading to chronic inflammation [28]. Consequently, ALP, AST, and ALT values were notably elevated, as is consistent with the literature. The TP levels in this study were also concordant with previous studies, showing higher values compared to the control group and a decrease in ALB levels, which is indicative of protein loss (proteinuria) [29]. This points to compromised Albumin/Globulin ratios resulting from both liver and kidney damage, which is further supported by the elevated urea and Crea levels. Given the slightly elevated levels of Chol and GGT, a potential issue with bile flow is suggested, which is also supported by the notably low levels of TB, warranting further investigation. Similar to the previous group, the AMYL, LPS, and TG levels were unexpectedly high, indicating the need for deeper investigation. Haematochemical modifications in anaplasmosis can include mildly elevated serum ALP and ALT activities [30,31]. However, our findings indicate a significant decrease in ALP values and an increase in ALT compared to the control group, possibly due to the presence of the infection without substantial stress or significant damage. Although hypoalbuminemia and proteinuria may occur without evidence of lower urinary tract disease [32], we did not observe any significant differences in our study group.

Notably, the group of dogs presenting a multiple-antibody response towards *Leishmania*, *Ehrlichia*, and *Rickettsia* showed higher values of TP, Glob, CPK, AST, ALT, and AMYL than the dogs that tested negative for all the pathogens investigated herein and the dogs that tested positive for a single pathogen. These results seem to suggest that multiple seropositivity could lead to greater liver function impairment and a greater inflammatory state in the positive animals. The detection of high values of these parameters during a clinical visit, even if during a simple routine screening, could alert the veterinarian to possible multi-infection by vector borne pathogens, especially in dogs coming from endemic areas.

## 5. Conclusions

In the present study, no molecular or parasitological diagnostic test was performed to confirm the presence of these infections, especially *Leishmania* and *Ehrlichia* infections, for which confirmation diagnostics are challenging. In fact, there are several studies and official recommendations endorsing that canine visceral leishmaniasis must be diagnosed by serological and parasitological or molecular tests or double different serological tests in order to discard false positive cases. Though the present study improves the knowledge currently available on the topic, further investigations are required in order to better clarify how co-infections affect haematochemical patterns in dogs living in endemic areas of VBDs. Moreover, the lack of molecular analysis, which can be considered a limitation of this study, did not allow us to exclude any cross-reactivity. The perturbation of such haematochemical parameters could suggest that the veterinarian perform an in deep investigation through molecular testing to confirm the identity of the pathogen.

## Figures and Tables

**Figure 1 vetsci-11-00205-f001:**
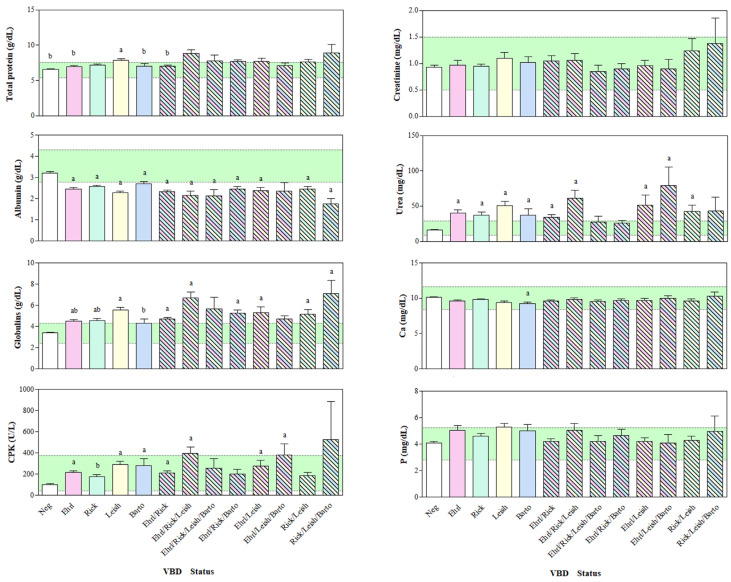
Mean values ± standard deviation (±SD) of Total Proteins, Albumin, Globulins, Urea, Creatinine, Creatine phosphokinase (CPK), Calcium (Ca), and Phosphorus (P) obtained from dogs divided into groups based on their seronegative (Neg) or simple seropositive status towards *Ehrlichia* (Ehrl), *Rickettsia* (Rick), *Leishmania* (Leish), *Borrelia* (Borr), *Anaplasma* (Anapl), and *Bartonella* (Barto), or their multiple seropositive status. The green area represents the reference range for each studied parameter. Significances (*p* < 0.05): a vs. Neg; b vs. Ehrl/Rick/Leish.

**Figure 2 vetsci-11-00205-f002:**
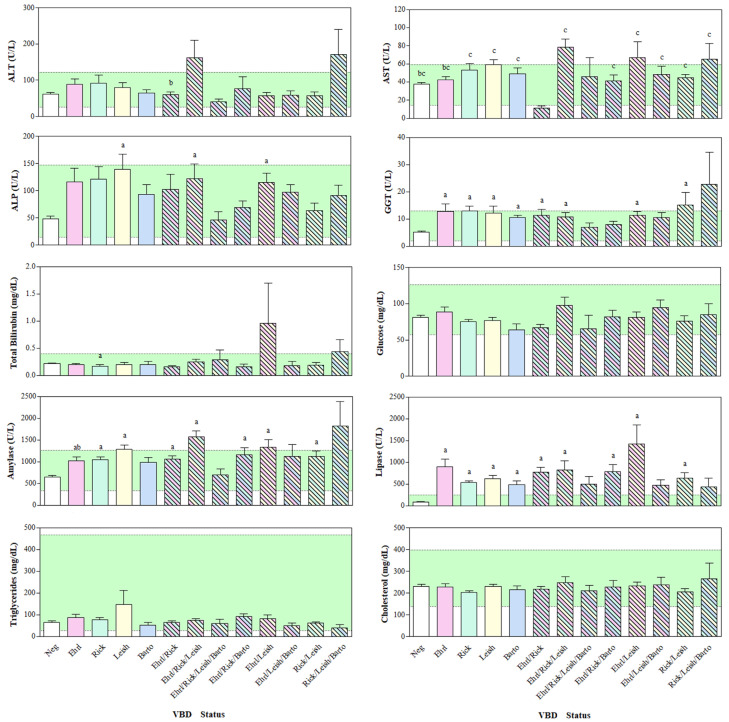
Mean values ± standard deviation (±SD) of Alanine Aminotransferase (ALT), Aspartate Aminotransferase (AST), Alkaline Phosphatase (ALP), Gamma-Glutamyl Transferase (GGT), Total Bilirubin, Glucose, Amylase, Lipase, Triglycerides, and Cholesterol obtained from dogs divided into groups based on their seronegative (Neg) or simple seropositive status towards *Ehrlichia* (Ehrl), *Rickettsia* (Rick), *Leishmania* (Leish), *Borrelia* (Borr), *Anaplasma* (Anapl), and *Bartonella* (Barto), their or multiple seropositive status. The green area represents the reference range for each studied parameter. Significances (*p* < 0.05): a vs. Neg; b vs. Ehrl/Rick/Leish; c vs. Ehrl/Rick.

**Table 1 vetsci-11-00205-t001:** The inclusion and exclusion criteria.

Inclusion	Exclusion
Owners informed consent for the scientific use of their animal’s data	Physical or historical signs of any kind
Dog clinically healthy on physical examination, free from external and internal parasites, and in good nutritional condition	Dog with history of vector-borne diseases
Biochemical profile checked at the time of hospital admission	Pharmacological treatment, including preventative flea/tick/mosquito treatments in the month prior to blood sampling
A serum aliquot stored at −20 °C	Dog vaccinated against the investigated diseases (i.e., leishmaniosis and borreliosis)

**Table 2 vetsci-11-00205-t002:** Signalment data for enrolled dogs divided into groups according to their response to serological testing.

Breed	Number of Dogs (%)	Age	Gender	Number of Dogs
Negative status (8.75%)	35 (8.75%)			
Mixed	11	Median 6 years	Males	22
German shepered	6	Range 2–14	Females	13
Yorkshire terrier	5			
English setter	3			
Boxer, Poodle, Rottweiler (two each)	6			
Beagle, Labrador retriever, Pitbull, Italian rough-haired segugio (one each)	4			
*Erlichia*-positive status	57 (14.25%)			
Mixed	25	Median 5 years	Males	26
Fonni’ Dog	10	Range 1–15	Females	31
German shepered, Italian rough-haired segugio (four each)	8			
Pitbull, Yorkshire terrier, Rottweiler, English setter (two each)	8			
Boxer, Pointer, Labrador retriever, Chihuahua, Dobermann, Jack russel terrier (one each)	6			
*Rickettsia*-positive status	75 (18.75%)			
Mixed	25	Median 5 years	Males	47
Fonni’ Dog	23	Range 1–15	Females	28
German shepered, Italian rough-haired segugio, (four each)	12			
Yorkshire terrier, Rottweiler (three each)	6			
Poodle, Dobermann, English setter (two each)	6			
Dachs hound, Cocker spaniel, Pitbull, (one each)	3			
*Leishmania*-positive status	51 (12.75%)			
Mixed	16	Median 4 years	Males	33
Fonni’ Dog	7	Range 1–13	Females	18
German shepered	5			
Italian rough-haired segugio, English setter (four each)	8			
Yorkshire terrier, Rottweiler, Boxer, Dobermann, Pitbull, Jack russel terrier, Labrador retriever, Pointer (two each)	8			
Poodle, American Staffordshire terrier, Dachs hound, Chihuahua, Cocker spaniel, Dalmatian, Shar-pei (one each)	7			
*Anaplasma*-positive status	1 (0.25%)			
Mixed	1	Median 5 years	Females	1
*Bartonella*-positive status	16 (4.00%)			
Fonni’ Dog	5	Median 4 years	Males	10
Mixed	4	Range 2–18	Females	6
German shepered, Italian rough-haired segugio (two each)	4			
Yorkshire terrier, Rottweiler, Dobermann (one each)	3			
Multiple seropositivity status	165 (41.25%)			
Mixed	65	Median 5 years	Males	86
Fonni’ Dog	32	Range 1–18	Females	79
German shepered	15			
Italian rough-haired segugio	15			
Yorkshire terrier	6			
English setter	5			
Boxer, Pitbull, Chihuahua, Labrador retriever, Pointer, Rottweiler (three each)	18			
Poodle, Dobermann, Dalmatian (two each)	6			
Dachs hound, Shih-tzu, Jack russel terrier (one each)	3			

## Data Availability

The data are contained within this article.

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
