# Peer review of "Haematochemical Profile of Healthy Dogs Seropositive for Single or Multiple Vector-Borne Pathogens"

_vetsci, 2024, doi:10.3390/vetsci11050205_

Round 1
Reviewer 1 Report
Comments and Suggestions for Authors
Although some of the items mentioned in my previous review were revised by the authors, the study was not approved by an ethics committee on animal use.
Plus, there are still some mistaken information in the Introduction section (e.g.: Line 43: both male and female ticks and fleas are bloodfeeders) and the authors did not discard or discuss the possibility of cross-reactivity for the serological tests.
That said, my recommendation is to reject the article.
Author Response
Dear Editor,
Thank you very much for reviewing the manuscript titled "Haematochemical profile of dogs tested seropositive to one or more canine vector-borne pathogens" (Manuscript ID: vetsci-2954778).
We were pleased to learn that the reviewer recommended the work for publication. We have diligently addressed the majority of the reviewer's concerns, as detailed below, and have made corresponding modifications to the manuscript.
As previously communicated via email, we have also updated the title to "Haematochemical profile of healthy dogs seropositive for single or multiple vector-borne pathogens". We believe this adjustment better reflects the content of the study.
We trust that the revisions we have made will render the manuscript suitable for publication.
We sincerely appreciate your assistance and valuable suggestions throughout this process. We eagerly await your feedback on our final version.
Thank you once again.
We have carefully considered his new suggestions. All answers to the specific comments are reported below.
Changes made in manuscript were highlighted in red.
Review Report Form (1)
Open Review
(x) I would not like to sign my review report
( ) I would like to sign my review report
Quality of English Language
( ) I am not qualified to assess the quality of English in this paper
( ) English very difficult to understand/incomprehensible
( ) Extensive editing of English language required
( ) Moderate editing of English language required
( ) Minor editing of English language required
(x) English language fine. No issues detected
|
Yes |
Can be improved |
Must be improved |
Not applicable |
|
|
Does the introduction provide sufficient background and include all relevant references? |
( ) |
( ) |
(x) |
( ) |
|
Are all the cited references relevant to the research? |
( ) |
(x) |
( ) |
( ) |
|
Is the research design appropriate? |
( ) |
( ) |
(x) |
( ) |
|
Are the methods adequately described? |
( ) |
( ) |
(x) |
( ) |
|
Are the results clearly presented? |
( ) |
(x) |
( ) |
( ) |
|
Are the conclusions supported by the results? |
( ) |
( ) |
(x) |
( ) |
Comments and Suggestions for Authors
Although some of the items mentioned in my previous review were revised by the authors, the study was not approved by an ethics committee on animal use.
- We thank Reviewer for the comment. We discuss this issue with Editor. In particular, we specified that “Regarding your question about the approval for the Ethical Information for the manuscript N° vetsci-2954778 entitled “Haematochemical profile of dogs tested seropositive to one or more
- canine vector-borne pathogens”, we confirm that the animals enrolled in the study have not been subjected to stressful conditions experimentally. The blood sampling was performed during the routine clinical examination of dogs, therefore, it does not need any separate ethical approval according to Italian legislation. The protocol of animal husbandry and experimentation were reviewed and approved in accordance with the standards recommended by the Guide for the Care and Use of Laboratory Animals and Directive 2010/63/EU for animal experiments. In particular, agreeing to the Guide for the Care and Use of Laboratory Animals and Directive 2010/63/EU for animal experiment, blood sampling procedures have been carried out according to the Section III of the cited Directive (Examples of different types of procedure assigned to each of the severity categories on the basis of factors related to the type of the procedure) point (b). The blood sampling was performed by trained veterinarian, while the owners were present. Moreover, each animal was enrolled in the study for free choice of the owners who have issued their informed consent.”
Plus, there are still some mistaken information in the Introduction section (e.g.: Line 43: both male and female ticks and fleas are bloodfeeders) and the authors did not discard or discuss the possibility of cross-reactivity for the serological tests.
-We thank Reviewer for the valuable comment. We corrected the mistake, and we specify the limitation of the study. We wrote “Though the present study improves the knowledge currently available on the topic, further investigations are advocated in order to better clarify how co-infections affect haematochemical pattern in dogs in dogs living in endemic areas for VBDs. Moreover, the lack of molecular analysis, which can be considered a limitation of the study, did not al-low us to exclude any cross-reactivity. the perturbation of such haematochemical param-eter could suggest veterinarian performing an in deep investigation by molecular testing in order to confirm the pathogen’s identity.”
That said, my recommendation is to reject the article.
Reviewer 2 Report
Comments and Suggestions for Authors
This manuscript describes the serological (and haematochemical) profiling of healthy dogs for several tick/ectoparasite borne microbes. The data provided in this manuscript is important for understanding the distribution/potential of ectoparasite borne diseases in companion canines, and for detailing the potential health impact of the disease(s) on the canines.
Introduction:
L41: Ehrlichia chaffeensis, Ehrlichia ewingii, and Ehrlichia muris can also be problematic for dogs.
L44: Do all of the aforementioned microbes follow gonadotrophic cycles? This should be clarified for each of the mentioned microbes.
L45-47: Both of these statements should be expanded/clarified to include additional relevant details.
L57-59: This statement should be expanded/clarified to include additional relevant details.
L64-65: This statement should be expanded/clarified to include additional relevant details.
L70-73: List the pathogens under evaluation.
Materials and Methods:
Section 2.1: Were any of the animals on preventative flea/tick treatments and for what duration of time? This detail should be included in the study.
L100: List the full name for the IFAT, followed by IFAT in parantheses.
L116-118: This sentence should be rewritten for clarification.
Serology: Is the cross-reactivity among the kits/species evaluated known? This is a detail that should be addressed in the manuscript.
L124: Double check proper capitalization.
L134-149: Briefly describe each procedure.
Results:
Table 2: List the age/gender for each individual canines (instead of grouping the data). The percentages (positive/negative) should be added to the Table. Symbols should be incorporated to designate the microbes associated with each co-infected animal.
L175-throughout Results: The data is very hard to understand/decipher using the A-F designations. The manuscript would be greatly improved if the pathogen names (or abbreviations of the names) were employed instead of the A-F designations.
Figures 1, 2: Consider increasing the size of the figures. As shown, they are very hard to read/interpret.
Discussion:
As written, a mix of full names and abbreviations are used for the metabolites/chemicals. This should be consistent throughout - use either full names OR abbreviations (not a mixture of both).
L225-276: This sections should be divided into paragraphs for each evaluated pathogen.
L277-281: The authors should consider employing molecular assays to confirm at least a subset of their data. Including molecular data in the results would greatly strengthen the manuscript.
Comments on the Quality of English LanguageEnglish adequate. Minor revisions only needed.
Author Response
Dear Editor,
Thank you very much for reviewing the manuscript titled "Haematochemical profile of dogs tested seropositive to one or more canine vector-borne pathogens" (Manuscript ID: vetsci-2954778).
We were pleased to learn that the reviewer recommended the work for publication. We have diligently addressed the majority of the reviewer's concerns, as detailed below, and have made corresponding modifications to the manuscript.
As previously communicated via email, we have also updated the title to "Haematochemical profile of healthy dogs seropositive for single or multiple vector-borne pathogens". We believe this adjustment better reflects the content of the study.
We trust that the revisions we have made will render the manuscript suitable for publication.
We sincerely appreciate your assistance and valuable suggestions throughout this process. We eagerly await your feedback on our final version.
Thank you once again.
We have carefully considered his new suggestions. All answers to the specific comments are reported below.
Changes made in manuscript were highlighted in red.
Review Report Form (2)
Open Review
(x) I would not like to sign my review report
( ) I would like to sign my review report
Quality of English Language
( ) I am not qualified to assess the quality of English in this paper
( ) English very difficult to understand/incomprehensible
( ) Extensive editing of English language required
( ) Moderate editing of English language required
(x) Minor editing of English language required
( ) English language fine. No issues detected
|
Yes |
Can be improved |
Must be improved |
Not applicable |
|
|
Does the introduction provide sufficient background and include all relevant references? |
( ) |
(x) |
( ) |
( ) |
|
Are all the cited references relevant to the research? |
(x) |
( ) |
( ) |
( ) |
|
Is the research design appropriate? |
( ) |
(x) |
( ) |
( ) |
|
Are the methods adequately described? |
( ) |
(x) |
( ) |
( ) |
|
Are the results clearly presented? |
( ) |
(x) |
( ) |
( ) |
|
Are the conclusions supported by the results? |
( ) |
(x) |
( ) |
( ) |
Comments and Suggestions for Authors
This manuscript describes the serological (and haematochemical) profiling of healthy dogs for several tick/ectoparasite borne microbes. The data provided in this manuscript is important for understanding the distribution/potential of ectoparasite borne diseases in companion canines, and for detailing the potential health impact of the disease(s) on the canines.
Introduction:
L41: Ehrlichia chaffeensis, Ehrlichia ewingii, and Ehrlichia muris can also be problematic for dogs.
-We thank Reviewer for the valuable comment. To the best of Authors knowledge, other Ehrlichia species (E. chaffeensis, E. ewingii and E. muris) have not been detected in dogs from Europe.
L44: Do all of the aforementioned microbes follow gonadotrophic cycles? This should be clarified for each of the mentioned microbes.
-We clarified this issue in the manuscript as Reviewer suggested.
L45-47: Both of these statements should be expanded/clarified to include additional relevant details.
We modified it.
L57-59: This statement should be expanded/clarified to include additional relevant details.
We modified it.
L64-65: This statement should be expanded/clarified to include additional relevant details.
We modified it.
L70-73: List the pathogens under evaluation.
We added the list the pathogens under evaluation.
Materials and Methods:
Section 2.1: Were any of the animals on preventative flea/tick treatments and for what duration of time? This detail should be included in the study.
-We thank Reviewer for the suggestion. Dogs subjected to pharmacological treatment including preventative flea/tick treatments in the last month prior to blood sampling were excluded from the study. We clarified this aspect in the text as well as in the table 1.
L100: List the full name for the IFAT, followed by IFAT in parantheses.
-Done.
L116-118: This sentence should be rewritten for clarification.
-We modified it.
Serology: Is the cross-reactivity among the kits/species evaluated known? This is a detail that should be addressed in the manuscript.
-We thank Reviewer for the valuable comment. The present study aimed to evaluate the antibody response towards the most common vector-borne pathogens and its impact on haematochemical profile in clinically healthy owned dogs living in Sardinia. Unfortunately, we did not perform molecular analysis, therefore, cross-reactivity cannot be excluded. However, it is well established that the serological method as indirect fluorescent antibody has become the standard testing method due to its simplicity, reliability and cot-effectiveness.
We added the limitation of the study in the text and we wrote “Though the present study improves the knowledge currently available on the topic, further investigations are advocated in order to better clarify how co-infections affect haematochemical pattern in dogs in dogs living in endemic areas for VBDs. Moreover, the lack of molecular analysis, which can be considered a limitation of the study, did not al-low us to exclude any cross-reactivity. the perturbation of such haematochemical parameter could suggest veterinarian performing a in deep investigation by molecular testing in order to confirm the pathogen’s identity.”
L124: Double check proper capitalization.
-Done
L134-149: Briefly describe each procedure.
-We thank Reviewer for the suggestion. We added the information as suggested.
Results:
Table 2: List the age/gender for each individual canines (instead of grouping the data). The percentages (positive/negative) should be added to the Table. Symbols should be incorporated to designate the microbes associated with each co-infected animal.
-We thank Reviewer for the suggestions. We added the percentage of positive/negative dogs as Reviewer required. However, it is hard to add information for each animal since a total of 400 dogs were enrolled in the study.
L175-throughout Results: The data is very hard to understand/decipher using the A-F designations. The manuscript would be greatly improved if the pathogen names (or abbreviations of the names) were employed instead of the A-F designations.
-We understand Reviewer’s concern. We modified the designations A-F with the abbreviations of the pathogen names.
Figures 1, 2: Consider increasing the size of the figures. As shown, they are very hard to read/interpret.
-We thank Reviewer for the comment. We improved the figures as suggested.
Discussion:
As written, a mix of full names and abbreviations are used for the metabolites/chemicals. This should be consistent throughout - use either full names OR abbreviations (not a mixture of both).
-We thank Reviewer for the comment. We modified them.
L225-276: This sections should be divided into paragraphs for each evaluated pathogen.
-Done.
L277-281: The authors should consider employing molecular assays to confirm at least a subset of their data. Including molecular data in the results would greatly strengthen the manuscript.
-We understand Reviewer’s concern. Unfortunately, we cannot perform molecular assays as we did not have samples.
Comments on the Quality of English Language
English adequate. Minor revisions only needed.
Reviewer 3 Report
Comments and Suggestions for Authors
The study provides valuable information on the hematochemical effects of dogs exposed to vector-borne diseases in an endemic region. In the reviewer's opinion, although correlations are observed between exposure to multiple pathogens and changes in hematochemical profiles, it would be beneficial to further explore the underlying mechanisms and confirm these findings using additional techniques such as molecular or parasitological testing. However, this does not prevent the publication of the work, which does not need current changes or modifications.
Author Response
Dear Editor,
Thank you very much for reviewing the manuscript titled "Haematochemical profile of dogs tested seropositive to one or more canine vector-borne pathogens" (Manuscript ID: vetsci-2954778).
We were pleased to learn that the reviewer recommended the work for publication. We have diligently addressed the majority of the reviewer's concerns, as detailed below, and have made corresponding modifications to the manuscript.
As previously communicated via email, we have also updated the title to "Haematochemical profile of healthy dogs seropositive for single or multiple vector-borne pathogens". We believe this adjustment better reflects the content of the study.
We trust that the revisions we have made will render the manuscript suitable for publication.
We sincerely appreciate your assistance and valuable suggestions throughout this process. We eagerly await your feedback on our final version.
Thank you once again.
We have carefully considered his new suggestions. All answers to the specific comments are reported below.
Changes made in manuscript were highlighted in red.
Review Report Form (3)
Open Review
( ) I would not like to sign my review report
(x) I would like to sign my review report
Quality of English Language
( ) I am not qualified to assess the quality of English in this paper
( ) English very difficult to understand/incomprehensible
( ) Extensive editing of English language required
( ) Moderate editing of English language required
( ) Minor editing of English language required
(x) English language fine. No issues detected
|
Yes |
Can be improved |
Must be improved |
Not applicable |
|
|
Does the introduction provide sufficient background and include all relevant references? |
(x) |
( ) |
( ) |
( ) |
|
Are all the cited references relevant to the research? |
(x) |
( ) |
( ) |
( ) |
|
Is the research design appropriate? |
(x) |
( ) |
( ) |
( ) |
|
Are the methods adequately described? |
(x) |
( ) |
( ) |
( ) |
|
Are the results clearly presented? |
(x) |
( ) |
( ) |
( ) |
|
Are the conclusions supported by the results? |
(x) |
( ) |
( ) |
( ) |
Comments and Suggestions for Authors
The study provides valuable information on the hematochemical effects of dogs exposed to vector-borne diseases in an endemic region. In the reviewer's opinion, although correlations are observed between exposure to multiple pathogens and changes in hematochemical profiles, it would be beneficial to further explore the underlying mechanisms and confirm these findings using additional techniques such as molecular or parasitological testing. However, this does not prevent the publication of the work, which does not need current changes or modifications.
-We thank Reviewer for the comments and suggestions. Moreover, we understand Reviewer’s concern on the need of further investigation in order to further explore the underlying mechanisms and confirm the results obtained in the current study. Unfortunately, we cannot perform molecular assays as we did not have samples. We specify the limitation of the study. We wrote “Though the present study improves the knowledge currently available on the topic, further investigations are advocated in order to better clarify how co-infections affect haematochemical pattern in dogs in dogs living in endemic areas for VBDs. Moreover, the lack of molecular analysis, which can be considered a limitation of the study, did not al-low us to exclude any cross-reactivity. the perturbation of such haematochemical param-eter could suggest veterinarian performing an in deep investigation by molecular testing in order to confirm the pathogen’s identity.”
Round 2
Reviewer 1 Report
Comments and Suggestions for Authors
Dear authors,
After the changes in the manuscript, it is more reasonable, clear and suitable for publication.
Reviewer 2 Report
Comments and Suggestions for Authors
Revisions adequate. Cheers to the authors!